# Glutathione during Post-Thaw Recovery Culture Can Mitigate Deleterious Impact of Vitrification on Bovine Oocytes

**DOI:** 10.3390/antiox12010035

**Published:** 2022-12-24

**Authors:** Lucia Olexiková, Linda Dujíčková, Alexander V. Makarevich, Jiří Bezdíček, Jana Sekaninová, Andrea Nesvadbová, Peter Chrenek

**Affiliations:** 1Agricultural and Food Centre (NPPC), Research Institute for Animal Production Nitra, Hlohovecká 2, 95141 Lužianky, Slovakia; 2Department of Botany and Genetics, Constantine the Philosopher University Nitra, Tr. A. Hlinku 1, 94974 Nitra, Slovakia; 3Department of Zoology, Faculty of Science, Palacký University Olomouc, 17. Listopadu 50, 77900 Olomouc, Czech Republic; 4Department of Biochemistry, Faculty of Science, Palacký University Olomouc, Šlechtitelů 27, 77900 Olomouc, Czech Republic; 5Institute of Biotechnology, Faculty of Biotechnology and Food Science, Slovak University of Agriculture in Nitra, Tr. A. Hlinku 2, 94976 Nitra, Slovakia

**Keywords:** bovine oocyte, vitrification, glutathione

## Abstract

Vitrification of bovine oocytes can impair subsequent embryo development mostly due to elevated oxidative stress. This study was aimed at examining whether glutathione, a known antioxidant, can improve further embryo development when added to devitrified oocytes for a short recovery period. Bovine in vitro matured oocytes were vitrified using an ultra-rapid cooling technique on electron microscopy grids. Following warming, the oocytes were incubated in the recovery medium containing glutathione (0, 1.5, or 5 mmol L^−1^) for 3 h (post-warm recovery). Afterwards, the oocytes were lysed for measuring the total antioxidant capacity (TAC), activity of peroxidase, catalase and glutathione reductase, and ROS formation. The impact of vitrification on mitochondrial and lysosomal activities was also examined. Since glutathione, added at 5 mmol L^−1^, significantly increased the TAC of warmed oocytes, in the next set of experiments this dose was applied for post–warm recovery of oocytes used for IVF. Glutathione in the recovery culture did not change the total blastocyst rate, while increased the proportion of faster developing blastocysts (Day 6–7), reduced the apoptotic cell ratio and reversed the harmful impact of vitrification on the actin cytoskeleton. These results suggest that even a short recovery culture with antioxidant(s) can improve the development of bovine devitrified oocytes.

## 1. Introduction

Vitrification of mammalian oocytes has wide possibilities of application in the field of assisted reproduction as well as preservation of the gene pool. However, the process of vitrification–warming is accompanied by considerable damage to the oocyte, which mostly results in oxidative stress [1,2,3,4,5]. Oxidative stress is a state arising when the generation of reactive oxygen species (ROS) by a cell or tissue exceeds the protective capacity of the intrinsic antioxidant mechanisms. Under physiological circumstances, the cells try to balance the production and disposal of ROS. Whilst physiological amounts of ROS are fundamental for cell function, the generation of supra-physiological ROS concentrations in a variety of situations, e.g., cryopreservation, can induce lipid peroxidation, DNA damage, and protein oxidation, resulting in organelle and plasma membrane injuries [5] and impaired oocyte functions [6].

Cryopreservation causes oxidative stress by different mechanisms, including osmotic stress, leading to increased oxidative metabolism within mitochondria [3] and mitochondrial injuries, which affect their function by increasing ROS production, depleting ATP, and triggering the intrinsic apoptotic pathway [4,7]. However, it is possible that other sources of ROS generation in oocytes outside the mitochondria are also involved. Oocytes are apparently rich in NADPH oxidase enzymes, and in some species strong lipoxygenase activity has been detected [8]. However, the contribution of such systems in the biology and pathology of the oocyte is currently unknown [6].

Despite the increasing amount of publication focusing on bovine oocyte vitrification, yields of transferable blastocysts after the in vitro fertilization of vitrified bovine oocytes remain low [9,10]. Oxidative stress has been suggested as a detrimental factor in fertilization success and the results could be improved, e.g., by the inclusion of antioxidants into the culture medium [11].

García-Martínez with colleagues [12] concluded that the addition of glutathione (GSH) to bovine oocytes during IVM prior to vitrification may be beneficial for embryo development presumably as a source of additional antioxidant protection. GSH is a tripeptide thiol (γ-glutamyl cysteinyl glycine) and is the prominent non-enzymatic part of defense against oxidative stress by reducing the sulfhydryl group. GSH is important for the elimination of xenobiotics and peroxide and performs an essential function in cell antioxidant protection [13]. However, oocytes and early embryos can synthesize only a limited amount of GSH, and GSH outside the cell cannot penetrate the oocyte membrane and enter the cells. Therefore, the acquisition of GSH at this stage depends on the cumulus cells absorbing thiols, which then synthesize GSH through the γ-glutamyl cycle [14].

Various other antioxidants (ascorbic acid, N-acetyl cysteine, melatonin, resveratrol, niacin or coenzyme Q10), similar to GSH, added to the culture have been shown to ameliorate the oxidative stress associated with oocyte vitrification in mouse [15,16], sheep [17], and cattle [12,18,19]. These researchers tried to increase the cryotolerance of oocytes by enhancing the antioxidant capacity of oocytes during in vitro maturation, mostly before freezing. In contrast, the goal of our work was to verify whether a relatively short recovery culture of oocytes post-warming in the presence of antioxidant protection could improve their developmental capacity by mitigating oxidative stress that occurred as a consequence of vitrification.

## 2. Materials and Methods

All the chemicals used in this study were purchased from Sigma-Aldrich Inc. (Saint-Louis, MO, USA) unless otherwise indicated.

### 2.1. Design of Experiments

In the first series of experiments (three replicates), vitrified oocytes after thawing were cultured during a 3-h recovery period in the maturation medium with 1.5 or 5 mmol L^−1^ glutathione (GSH) or without GSH (30 oocytes per group). Subsequently, the oocytes were lysed in a lysis buffer for measuring the total protein concentration and following antioxidant parameters: total antioxidant capacity, activity of peroxidase, catalase and glutathione reductase. A group of fresh oocytes was also lysed and served as a control. Based on the results of these measurements, a proper dose of GSH for further experiments was selected.

In the second series of experiments, both vitrified–warmed and fresh oocytes were stained to assess the activity of mitochondria (MitoTracker Green) and lysosomes (LysoTracker DeepRed). A part of the vitrified–warmed oocytes was cultured with or without GSH (5 mmol L^−1^) to quantify the formation of ROS.

In the third series (6 repetitions), vitrified–warmed oocytes during the recovery period were cultured without (VGSH− group) or with GSH (5 mmol L^−1^; VGSH+ group). Non-vitrified (fresh) oocytes served as a control. Subsequently, both groups were in vitro fertilized and cultured to the blastocyst stage. Cleavage rate and blastocyst development (day 6−8) were evaluated. Randomly selected blastocysts were fixed and stained to evaluate the total number of cells (DAPI), the proportion of apoptotic cells (TUNEL-assay), and the quality of actin cytoskeleton (Phalloidin-TRITC).

### 2.2. Oocyte Retrieval and In Vitro Maturation (IVM)

The source of the ovaries were cows from a local slaughterhouse. Immediately after slaughter, the ovaries were brought at a temperature of 22–25 °C. In the laboratory, cumulus-oocyte complexes (COCs) were aspirated from antral follicles (2–8 mm in size) with a sterile 5 mL syringe and needle. The obtained COCs were washed several times in the holding medium (E-199 with HEPES and 10% foetal bovine serum—FBS). Oocytes with a complete and well-adjacent cumulus cell layer and homogeneous medium brown ooplasm were selected for in vitro maturation (IVM). The medium used for maturation was E-199 (Gibco) with L-alanyl glutamine, to which we added sodium pyruvate (0.25 mmol·L^−1^), gentamicin (50 mg·mL^−1^), 10% FBS, and FSH/LH (1/1 IU., Pluset). Oocyte vitrification followed IVM lasting 21 h, at 38.5 °C and 5% CO_2_ in the atmosphere. After several weeks, the COCs were warmed and cultured for 3 h either with or without glutathione for post-thaw recovery. For the control group, fresh oocytes were matured in the same IVM medium for 24 h.

### 2.3. Cryopreservation of Oocytes

Vitrification of oocytes was carried out according to the methodology previously described in Olexiková et al. [20]. Briefly, oocytes after 21 h of maturation were briefly (30 s) vortexed at high speed to partially remove expanded cumulus cells. Partially cleaned oocytes were equilibrated in an equilibration solution (ES: 3% ethylene glycol (EG) in E199-HEPES, supplemented with 10% FBS) for 12 min. Then, the oocytes were vitrified. The vitrification solution (VS) was composed of 30% EG with 1 M sucrose in E199-HEPES with 10% FBS. Oocytes were exposed to VS for 25 s at room temperature. A small group of oocytes, usually containing 15–17 pcs, was then placed together on an electron microscopic grid (300 mesh, Nickel) lined with filter paper. The VS was immediately aspirated with filter paper and the mesh with the oocytes was immersed in liquid nitrogen. Grids were stored in liquid nitrogen for several weeks. The devitrification procedure was as follows. After being removed from liquid nitrogen, the grids with oocytes were immediately immersed in a pre-heated (37 °C) warming solution (0.5 M sucrose in M199–HEPES) for 1 min. Subsequently, three dilution solutions with gradually decreasing sucrose concentration in the holding medium (0.25 M, 0.125 M and 0.0625 M) were used. Each wash lasted 3 min. Finally, warmed oocytes were washed with holding medium without sucrose. Oocytes that did not survive vitrification/warming (shrunk dark ooplasm, ruptured *zona pellucida*) were excluded from further experiments.

### 2.4. Antioxidant Capacity and Activity of Antioxidant Enzymes in Oocytes

The activities of the studied enzymes (catalase, peroxidase, and glutathione reductase), protein concentration, and total antioxidant capacity (TAC) were determined spectrophotometrically using a Synergy microplate reader (BioTek, USA). Samples of fresh or vitrified–warmed oocytes with or without GSH (30 oocytes for each group) were lysed in 50 μL of lysis buffer mixture (Protease inhibitor cocktail, Abcam) and stored at −20 °C until analyses. Prior to other analyses, the protein concentration in the samples was determined by Bradford’s method using bovine serum albumin as a standard.

The total antioxidant capacity (TAC) of the samples was determined by the ABTS (2,2-azinobis (3-ethylbenzthiazoline-6-sulfonic acid)) method [21], which is based on the ability of the substances to quench the ABTS cation radical. The reaction mixture (200 μL) consisted of 50 μL of oocyte lysate diluted in 0.1 M potassium phosphate buffer (pH 7.0) with 0.5 mM of ammonium persulfate (APS) and 0.8 mmol L^−1^ of ABTS. After 25 min of incubation at room temperature, the decrease in absorbance at 734 nm was measured and the total antioxidant capacity was compared with the antioxidant capacity of Trolox, which served as a standard.

The catalase activity in the samples was determined spectrophotometrically according to Rahiminejad [22]. The reaction mixture (230 μL) contained 10 μL of oocyte lysate in 0.1 M potassium phosphate buffer (pH 7.0). The reaction was started by the addition of hydrogen peroxide (final concentration—12 mmol L^−1^). The decrease in absorbance caused by the decrease in the concentration of hydrogen peroxide in the reaction by the catalytic action of the enzyme was recorded at a wavelength of 240 nm at 30 °C for 5 min.

The peroxidase activity was determined spectrophotometrically according to Lyttle and DeSombre [23]. The reaction mixture (175 μL) contained 10 μL of oocyte lysate in 0.1 M potassium phosphate buffer (pH 6.0) and 0.113 M guaiacol. The reaction was started by the addition of hydrogen peroxide (final concentration—10 mmol L^−1^). Color product formation was measured at absorbance of 436 nm, at 30 °C for 1 min.

Glutathione reductase (GR) activity was determined by a spectrophotometric method, based on measuring the decrease in absorbance, caused by nicotinamide adenine dinucleotide phosphate (NADPH) oxidation [24]. Two substrates, glutathione disulphide (GSSG) and NADPH, were used in this assay and the rate of decrease in absorbance was directly proportional to GR activity. The reaction mixture (280 μL) contained 20 μL of oocyte lysate in 0.1 M potassium phosphate buffer (pH 7.4), 0.16 mmol L^−1^ of NADPH and 1 mmol L^−1^ of GSSG. The decrease in absorbance at 340 nm was recorded at 25 °C for 5 min.

### 2.5. Evaluation of Mitochondrial and Lysosomal Status in Oocytes

Mitochondria and lysosomes in oocytes were detected by simultaneous staining with MitoTracker^TM^ Green (Invitrogen, MA, USA) and LysoTracker^TM^ Deep Red (Invitrogen, MA, USA), respectively, in order to evaluate fluorescent activity and distribution in fresh and vitrified oocytes. Intact (unfixed) oocytes were incubated in LysoTracker solution (50 nmol L^−1^) for 30 min at 37 °C, washed three times in a phosphate–buffered saline (PBS) with 0.6% of polyvinylpyrrolidone (PVP) and subsequently incubated in MitoTracker solution (200 nmol L^−1^) for 30 min at 37 °C, according to the procedure of Jeseta [25]. Afterwards, oocytes were washed in PBS–PVP, fixed with 4% neutrally buffered formalin for 20 min at room temperature, washed again, covered with a drop of Vectashield anti–fade mounting medium with DAPI (Vector Laboratories, Burlingame, CA, USA), and mounted onto glass slides with a coverslip.

Stained oocytes were scanned by an LSM 700 Zeiss confocal laser scanning microscope (Carl Zeiss Slovakia, s.r.o., Bratislava, Slovak Republic) and the mitochondrial and lysosomal activities were evaluated based on fluorescence intensity using ImageJ software. The corrected total cell fluorescence (CTCF = integrated density − (area of selected cell x mean fluorescence of background readings) formula was used to determine the fluorescence value. Mitochondrial and lysosomal distribution was evaluated based on occurring patterns (aggregated, diffused, and mixed). In the case of an aggregated pattern, the numbers of aggregates per oocyte were counted and rates of co-localization of mitochondria and lysosomes were evaluated.

### 2.6. Assay of ROS Formation in Oocytes

The level of intracellular ROS was assessed by fluorescent staining using CellROX Green (Invitrogen, MA, USA). Denuded oocytes were incubated for 30 min in a dye mixture (according to the producer´s guide), washed in PBS–PVP, fixed for 10 min in 4% of neutrally buffered formalin, covered with a drop of Vectashield anti-fade mounting medium with DAPI, and mounted onto glass slides with a coverslip. Stained oocytes were scanned by an LSM 700 Zeiss confocal laser scanning microscope, and the ROS formation was evaluated based on fluorescence intensity using ImageJ software and the formula described above.

### 2.7. In Vitro Fertilization (IVF) of Vitrified-Warmed Oocytes and Embryo Culture

Morphologically good-looking oocytes from both control and vitrified/warmed groups intended for IVF were washed in IVF–TALP medium (Tyrode–Albumin–Lactate–Pyruvate solution, 10 mg mL^−1^ heparin, 50 mg mL^−1^ gentamicin) and put into 100 µL droplets of IVF–TALP medium previously covered with a mineral oil into Nunc 4-well plates (ThermoFisher Scientific, Bratislava, Slovak Republic). Afterwards, the sperm (at 2 × 10^6^ per mL) and PHE solution (20 mmol L^−1^ penicillamine, 10 mmol L^−1^ hypotaurine, 1 mmol L^−1^ epinephrine) were added and incubated at 38.5 °C with 5% CO_2_ in atmosphere. At 18 h from the start of IVF, oocytes (now presumptive zygotes) were cleared of adhered spermatozoa and cumulus cell debris by vortexing in a microtube containing 75 µL of holding medium for 30 s. The zygotes were subsequently placed in prepared 4-well culture dishes (NUNC) with a grown confluent monolayer of BRL feeder cells (BRL-1, EACCC). A medium prepared according to composition of Menezo B2, supplemented with 10% FBS, was used for cultivation. The development of in vitro fertilized oocytes was subsequently observed up to the blastocyst stage. We evaluated cleavage rate on Day 2 after fertilization and blastocyst rate on Days 6, 7 and 8.

### 2.8. Fluorescent Staining of Blastocysts

The total number of cells in blastocysts, indicating their proliferative capacity, was counted based on blue fluorescent staining (DAPI) of their nuclei. In addition to the total number of cells, the incidence of apoptotic cells in blastocysts was monitored by the TUNEL method, as well as the quality of the actin cytoskeleton after Phalloidin-TRITC staining. For staining, blastocysts were briefly washed in PBS-PVP solution (0.6%). They were fixed with freshly prepared neutral buffered formalin (4%) for 10 min. The In Situ Cell Death Detection Kit (Roche Slovakia, Bratislava, Slovak Republic) was used to mark apoptotic cells according to the manufacturer’s instructions. Briefly, the procedure was run after permeabilization in the reaction mixture (TdT buffer, FITC-dUTP, and TdT) for one hour at a temperature of 37 °C in incubator. After completion, blastocysts were washed in cold PBS-PVP solution.

Afterwards, blastocysts were incubated in phalloidin-TRITC solution at room temperature for 30 min to stain the actin cytoskeleton. Then, the embryos were washed in PBS–PVP solution and on a glass slide covered with a drop of Vectashield anti-fade medium containing DAPI fluorochrome, with the help of which microscopic preparations were made. They were wrapped in opaque packaging and stored in the refrigerator until analysis. Fluorescence analyses of blastocysts were performed using a Leica fluorescence microscope and an LSM 700 Zeiss confocal laser scanning microscope.

The actin cytoskeleton was classified according to Makarevich [26] with slight modification, on the basis of the appearance of actin filaments in blastocysts. Good quality actin was considered as sharply stained actin filaments of reticular shape in cell borders, or actin staining with minor deviations (blastomeres with slightly less pronounced actin filaments, without sharp borders on membranes). Poor quality actin was characterized by large areas lacking actin staining or visible actin largely aggregated into intracytoplasmic clumps.

### 2.9. Statistics

Data on embryo development rates of control and vitrified/warmed bovine oocytes are represented by 6 replicates. Distribution to stages according to cleavage and blastocyst formation and distribution according to actin quality were analyzed by a Pearson´s Chi-square statistic test with the Yates correction for continuity. The incidence of apoptotic cells was represented as a percentage value (index) from the total cell number. All indexes before statistical analysis were subjected to arcsine square root transformation. The normality of the data was tested by Shapiro–Wilk test and equal variance test. The differences in the mean value of experimental versus control group in the total number of cells, the incidence of apoptotic cells and mean values of total protein and enzymes activity were evaluated by Student’s *t*-test. All pairwise multiple comparison procedures and one-way ANOVA were conducted following the Holm–Sidak method. The overall significance level used was 0.05. All statistical analyses were performed using SigmaPlot 11.0 (Systat) software.

## 3. Results

### 3.1. Effect of Vitrification and GSH on Antioxidant Capacity and Activity of Antioxidant Enzymes in Oocytes

In order to determine the effect of vitrification and the addition of GSH after devitrification on important components of oocyte antioxidant protection, we investigated the total antioxidant capacity, activity of catalase, peroxidase, and glutathione reductase in lysed cumulus-oocyte complexes. Total protein content did not differ between individual samples (2.19, 1.99, 2.11 and 2.01 mg/mL for control, VGSH−, VGSH 1.5 and VGSH 5, resp.; VGSH−—vitrified oocytes without GSH supplementation; VGSH+—vitrified oocytes with GSH supplementation 1.5 or 5.0 mmol L^−1^) indicating the uniformity of all tested groups. Vitrification of oocytes led to an increase in peroxidase activity and a decrease in catalase activity (*p* < 0.05). The addition of GSH at both concentrations to devitrified oocytes significantly reduced catalase activity. Oppositely, the addition of 5 mmol L^−1^ GSH significantly increased the activity of peroxidases and, at the same time, the total antioxidant capacity was increased in this group (VGSH 5) compared to the other groups (Figure 1). Neither vitrification nor the addition of GSH significantly affected the activity of glutathione reductase.

### 3.2. Effect of Vitrification on Mitochondrial and Lysosomal Status in Oocytes

Vitrification caused a significant decrease in mitochondrial activity measured as relative fluorescence intensity after fluorescent staining. Vitrification also caused a significant decrease in the intensity of lysosomal fluorescence (Figure 2).

Most of the oocytes showed a mixed (diffused and aggregated) distribution pattern of mitochondria (Figure 3C). Mitochondria were distributed rather diffusely with the appearance of several aggregates in 95% of control oocytes (41 out of 43) but only in 76% of vitrified oocytes (36 out of 47). In the vitrified group, 15% of oocytes (seven out of 47) contained only mitochondria arranged in aggregates (Figure 3B). Less than 10% of oocytes in both groups had only diffused mitochondria in the ooplasm (Figure 3A).

In the case of lysosomes, the aggregated pattern of distribution prevailed in both control (84%; 36 out of 43) and vitrified (89%; 42 out of 47) oocytes (Figure 4). More than 80% of lysosomal aggregates were co-localized with mitochondrial aggregates (Figure 4).

### 3.3. Effect of Vitrification and GSH on ROS Production in Oocytes

Taking into account the results of the previous experiments, we used only a concentration of 5 mmol L^−1^ to test the effectiveness of GSH in the next experiments. Vitrification caused a significant increase in the production of ROS in oocytes, which was visible as increased green fluorescence intensity (Figure 5A–F).

GSH added to the post-warming recovery culture at 5 mmol L^−1^ significantly reduced the ROS production in devitrified oocytes (Figure 6).

### 3.4. Effect of Oocyte Vitrification and GSH on In Vitro Embryo Development

In this experiment, the developmental potential of the vitrified–warmed oocytes was compared to those exposed to 5 mmol L^−1^ GSH during the recovery period and to the control oocytes (non–vitrified). In our study, a total of 840 oocytes were used, of which 421 oocytes were vitrified. Devitrified oocytes after post–warming recovery (3 h) were fertilized in vitro. This experiment was performed in six replicates.

The cleavage rate in individual experiments varied: from 50.05% to 69.81% in vitrified oocytes, from 47.05% to 72% in the VGSH+ group, and from 59.09% to 78.18% in the control oocytes. Blastocyst rate varied from 13.04% to 29.41% for vitrified oocytes, from 9.52% to 24.44% for VGSH+, and from 27.08% to 49.09% for the control oocytes. Average values of cleavage and blastocyst rate of in vitro fertilized oocytes are presented in Table 1.

The cleavage rate of fertilized zygotes did not differ among vitrified and control groups irrespectively of GHS addition (*p* > 0.05). However, both vitrified groups (irrespective of GSH) had significantly lower total blastocyst rates compared to the control group (*p* < 0.05). Taking into consideration fast developing blastocysts (Day 7) from total blastocyst rate (TBR), GSH addition significantly increased the D7 blastocyst rate (71.79%) compared to the GSH-free group (55.00%; *p* < 0.01) up to the level of non-vitrified control (74.75%).

In order to determine the dynamics of embryo development in the tested groups, all obtained blastocysts were assigned to the day when the blastocyst stage reached from 6th to the 8th day (Table 2).

Most of fast developing (D6 and D7) blastocysts were recorded in the vitrified group supplemented with GSH, which was significantly higher than in the vitrified group without GSH (*p* < 0.01) and was comparable to control oocytes. The most of slow-developing (D8) blastocysts were recorded in the vitrified group without GSH supplementation compared to GSH-stimulated vitrified (*p* < 0.01) or control (*p* < 0.01) groups.

### 3.5. Effect of Vitrification and GSH on Blastocyst Quality

Blastocyst quality was evaluated based on three parameters: average blastocyst cell number, which reflects the intensity of proliferation, the incidence of apoptotic (TUNEL-positive) cells, and the quality of actin cytoskeleton (Figure 7).

Vitrification significantly decreased the total cell number of blastocyst produced from devitrified oocytes (98.35) and GSH did not significantly improve this parameter (101.00) compared to control (112.56). Vitrification has no effect on the incidence of apoptotic cells (TUNEL–index) in the blastocysts. Nevertheless, the addition of GSH to the recovery culture even suppressed apoptosis in this group of blastocysts compared to control (*p* < 0.05; Table 3).

Vitrification significantly lowered the percentage of blastocysts with good quality actin (Figure 8A) and increased the number of embryos with poor quality actin (Figure 8B) compared to the control (Table 4). GSH addition resulted in elevating the percentage of blastocysts with the good actin cytoskeleton (75.00%) up to a level even slightly higher than in the control group (71.43%).

## 4. Discussion

Vitrification of bovine oocytes still results in a significantly lower percentage of blastocysts after their fertilization and culture as compared to fresh embryos. Several authors studied the causes and possibilities to increase the efficiency of in vitro embryo production from thawed oocytes in bovine as well as in other animal species and humans (for review see [10,27,28]).

Low fertilization rates of cryopreserved oocytes are reportedly associated with chilling and freezing injuries including membrane damage, zona hardening, due to premature release of cortical granules or abnormal increase of cytoplasmic free calcium ions [29], spindle disorganization and loss or clumping of microtubules [30,31]. It is also obvious that some epigenetic abnormalities in embryo development may occur after oocyte vitrification, which may be explained by altered expression of genes associated with epigenetic regulations [32]. It is also possible that oocyte vitrification alters global methylation in resultant embryos, although such alteration in the oocytes was not detected [33].

It was demonstrated that oocyte vitrification can disturb the reduction–oxidation (redox) status, reduce glutathione content and increase ROS levels, resulting in the damage of biomolecules such as DNA, proteins and membrane lipids and leading to mitochondrial dysfunction [34,35]. Mitochondrial dysfunction induced by cryopreservation can be mitigated after thawing, when the damaged mitochondria are removed and de novo synthesis occurs to restore the function of mitochondria [36].

In our work, we investigated the response of some important cell organelles, such as mitochondria to damage caused by oocyte vitrification. Mitochondrial activity was significantly decreased in the vitrified group compared to the control, what indicates the mitochondrial deficiency of oocytes after vitrification. A similar decrease in mitochondrial activity was described in vitrified mouse oocytes [37,38]. The negative impact of vitrification on mitochondrial function was described in bovine oocytes by Zhao [39]. The decrease in mitochondrial membrane potential as well as changes in mitochondrial distribution were also observed in vitrified porcine oocytes [4]. Similarly, in our study, slight changes in the distribution of mitochondria in the vitrified/warmed oocytes were revealed.

In lysosomes, the aggregated pattern of distribution prevailed in both vitrified and fresh groups (83.72–89.36%) and most of the mitochondrial and lysosomal aggregates were co-localized. High co-localization of mitochondrial lysosomal aggregates (more than 80%) indicates an interconnection of these two organelles and the importance of rapid removal of damaged mitochondria by organelles with lysosomal activity. The decrease in the fluorescence intensity of lysosomal organelles in vitrified/warmed oocytes is probably due to the increased involvement in macrophagic/autophagic processes within the repair of cryodamages. In this process, autophagosomes are directed towards lysosomes, where their outer membrane fuses with the surrounding membrane to release single-membrane bound structures (autophagic bodies) into the lumen. The acidic environment inside the lysosome promotes the digestion of macromolecules by hydrolysis [40]. One of the considered possible hypotheses is that, during this process, a “dilution” of the strongly acidic content occurs, which probably leads to a decrease in the intensity of the LysoTracker fluorescence, but it needs further investigation.

Interestingly, additional post-thaw incubation of oocytes resulted not only in the recovery of the meiotic spindle [41], but also mitochondrial function [42,43] compared to their status immediately after warming. Iwata [36] in his study stated that antioxidants can help in mitochondrial recovery from cryopreservation–induced damage. In view of these facts, it is reasonable to expect that the addition of proper biological substances during the recovery period can further reverse the damage initiated by vitrification and provide better embryo development. Therefore, we tested the possibility of using the addition of glutathione during the recovery period (3 h) to alleviate the increased oxidative stress caused by vitrification/warming.

Glutathione is known as an effective substance, directly or indirectly, in many important biological processes including the synthesis of proteins and DNA, transport, enzyme activity, metabolism and protection of cells [44]. Glutathione occurs in the cell in reduced (GSH) and oxidized (GSSG) forms, maintains the redox balance in the cell, and serves to regenerate the reduced form of other antioxidants. Alterations in GSH content during oocyte vitrification may affect embryo cleavage by preventing the formation of the male pronucleus [45]. Trapphoff et al. [15] concluded that increased GSH content in mouse oocytes improved their cryotolerance. García-Martínez et al. [12] confirmed that GSH-ethyl-ester protects bovine in vitro maturing oocytes against oxidative stress induced by subsequent vitrification/warming.

Although low and controlled ROS level is adequate for normal cell function, the vitrification process substantially increases ROS level in oocytes. An imbalance between the production and neutralization of ROS causes oxidative stress in the oocytes with a consequent impact on developmental competence [46]. In our study, the detection of ROS confirmed the increased oxidative stress of vitrified–warmed bovine oocytes, and GSH supplementation during post thaw recovery mitigated the consequences of vitrification by suppressing the elevated ROS activity in vitrified oocytes. Similar indications also appeared in earlier reports in bovine [12,35] and in porcine [2] oocytes. In mouse oocytes, significantly lower ROS activity was detected in the vitrified group supplemented with glutathione-ethyl-ester in the IVM medium compared to the vitrified group without supplementation [15].

In vitrified oocytes, we observed also changes in the most important intracellular enzymatic antioxidant systems, which may be the consequences of oxidative stress. The enzyme activity of bovine oocytes in response to oxidative stress during in vitro maturation was previously studied by Cetica [47]. In our experiments, vitrification induced the activity of peroxidases in warmed oocytes. This induction could be related to a higher level of peroxidation products as a result of increased oxidative stress. The increasing activity of peroxidases was associated with the increasing GSH dose in the post-thaw recovery medium. GSH plays a role in the functioning of enzymatic antioxidant systems, since glutathione peroxidases, which are active with both hydrogen peroxide and organic hydroperoxide substrates, use GSH as an electron donor. Therefore, improving accessibility of GSH may be the reason for the highly elevated peroxidase activity [48]. Elevated peroxidase activity (at 5 mmol L^−1^ of GSH) was also associated with a significant increase in the total antioxidant capacity. However, the GSH induction was not observed equally for all enzymatic antioxidant systems.

Particularly, catalase activity in oocytes was reduced after vitrification–warming and GSH was able to suppress it more. Catalase in organisms has two enzymatic activities depending on the concentration of hydrogen peroxide. If the concentration is high, catalase removes hydrogen peroxide by forming H_2_O and O_2_. However, at a low concentration of hydrogen peroxide and in the presence of a suitable hydrogen donor, catalase is otherwise regulated. Catalase can act peroxidically, removing hydrogen peroxide but oxidizing its substrate [49]. It is not clear, why catalase activity in oocytes decreases after thawing. From previous findings on the functioning within the reproductive tract, it is known that, during folliculogenesis, catalase activity decreases along with an increase in total antioxidant capacity [50]. Catalase also plays an important role in the protection of spermatozoa during their journey in the oviduct [51]. Possible causes of catalase inhibition may be regarding to elevated peroxidases in the substrate or product regulation by negative feedback.

Although in our experiments, the addition of GSH to the recovery culture did not have a significant effect on the blastocyst rate compared to the vitrified group without GSH, it had an effect on the speed, at which the embryos developed into the blastocyst. In our previous study [20], we already noted that the in vitro production of embryos from vitrified oocytes resulted in a slowing down of development to the blastocyst stage. Similarly, other authors have noted the association of oocyte cryopreservation with cell division delay [52,53]. The morphokinetics and speed of development of pre-implantation embryos are one of the important parameters related to their development potential and quality. Studies in cattle have linked early developmental kinetics and blastocyst formation with their viability [54,55]. Furthermore, caspase activity was significantly higher in slow developing embryos in comparison with fast cleavers, which may be related to increased induction of apoptosis [56]. Moreover, those blastocysts that form early appear to have a greater likelihood of providing live offspring after transfer [57]. New studies also revealed different metabolic profiles and expression patterns in different rapidly developing bovine embryos [58,59]. Therefore, the faster development of blastocysts in the VGSH+ group can be considered a positive effect of glutathione on vitrified oocytes. The developmental pattern in this group did not differ from that in the fresh oocyte group.

Total cell and apoptotic number are important parameters of embryo development and viability. Several studies described lower cell number [60,61] and a higher ratio of apoptotic nuclei [35] in blastocysts from vitrified oocytes compared to fresh ones. Our results are in concert with these reports, as the total cell number in the vitrified group (98.35 ± 6.25) was less than in the control (112.56 ± 3.39). GSH addition did not affect this value (101 ± 4.58). However, GSH significantly decreased TUNEL-index in embryos from vitrified oocytes compared to the control group (8.18% vs. 11.17%), which confirms the anti-apoptotic effect of GSH during the recovery culture. Similar results were obtained by Rocha-Frigoni et al. [50] using intracellular (cysteine and β-mercaptoethanol) and extracellular (catalase) antioxidant supplementation at different times during IVP on bovine embryos after vitrification/re-expansion. Although re-expansion rates were not affected by the treatments, ROS levels and apoptotic incidence were lower in these groups compared to the untreated control.

The actin cytoskeleton plays an important role in development, as its ability is to coordinate cellular functions and integration of various signals [62]. During vitrification, the actin cytoskeleton can be affected, which can result in poor embryo development. No irregular patches or clumps were detected in the actin cytoskeleton of mouse embryos vitrified at various developmental stages using the modified droplet vitrification method [63]. However, Dalcin et al. [64] in their study observed many points of cytoskeleton disruption in vitrified sheep embryos. Our results indicate that the percentage of embryos with a good quality actin cytoskeleton was reduced in the vitrified group (64.70%). However, GSH increased the proportion of embryos with good quality actin (75.00%), which was comparable to the control group (71.43%), indicating a trend of GHS to improve the quality of actin cytoskeleton.

## 5. Conclusions

According to our results, it is obvious that GSH confirmed its protective role against damage induced by vitrification. Even a relatively short period of post-thaw recovery culture (3 h) represents a space for improving the development of embryos produced in vitro from vitrified/warmed bovine oocytes. GSH, added at 5 mmol L^−1^, significantly increased the total antioxidant capacity of warmed oocytes. Although the addition of GSH into the recovery culture did not increase the blastocyst rate, it was able to induce faster blastocyst formation and improve their quality by reducing the incidence of apoptotic cells. These results expand previous knowledge about the mitigating effect of antioxidants on the harmful consequences of the cryopreservation of oocytes in various experimental designs.

## Figures and Tables

**Figure 1 antioxidants-12-00035-f001:**
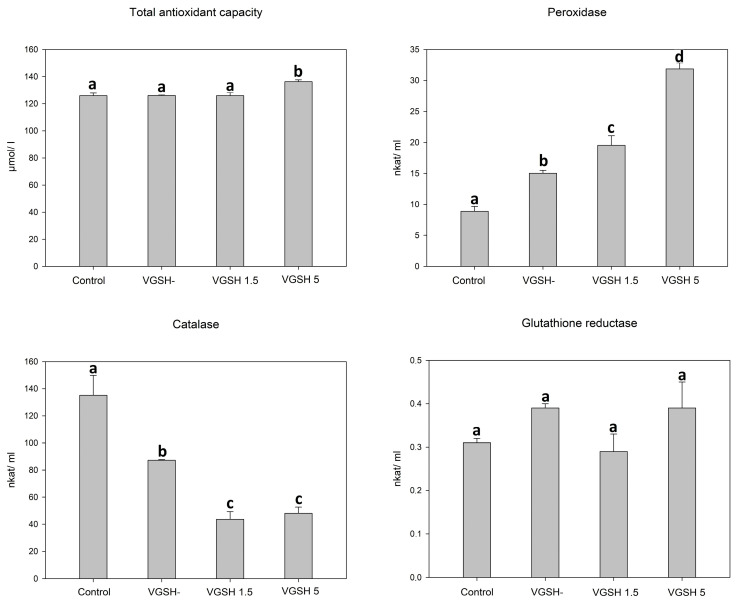
Effect of vitrification and different GSH doses on the antioxidant characteristics of oocytes. VGSH−—vitrified oocytes without GSH supplementation; VGSH+—vitrified oocytes with GSH supplementation (1.5 or 5.0 mmol L^−1^). a versus b versus c versus d—differences are significant at *p* < 0.05.

**Figure 2 antioxidants-12-00035-f002:**
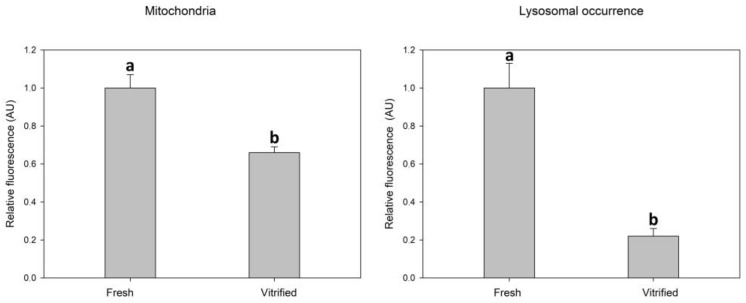
Effect of vitrification–warming of oocytes on mitochondria and lysosomes relative fluorescence intensity. Values of relative fluorescence intensity are in arbitrary units (AU). Differences among vitrified and fresh oocyte groups are significant at *p* < 0.05.

**Figure 3 antioxidants-12-00035-f003:**
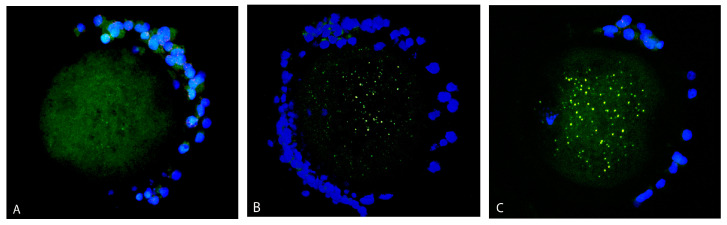
Distribution pattern of mitochondria (green signal) in oocytes: diffused distribution (**A**), aggregated (**B**) and mixed (diffused and aggregated) distribution pattern of mitochondria (**C**). Cumulus cells are stained by DAPI (blue signal).

**Figure 4 antioxidants-12-00035-f004:**
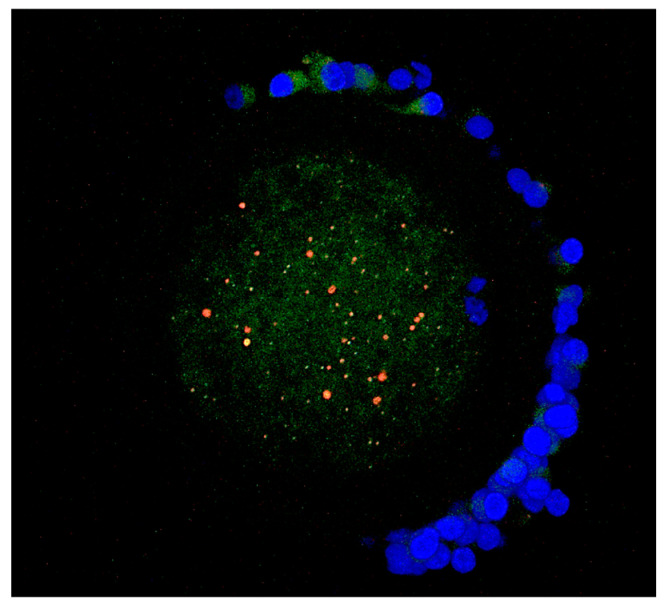
Aggregate co–localization of mitochondria (green fluorescence) and lysosomes (red fluorescence) in a merged image. For closely located organelles, the overlapping of red and green creates a yellow-orange color. Chromatin are stained by DAPI (blue fluorescence).

**Figure 5 antioxidants-12-00035-f005:**
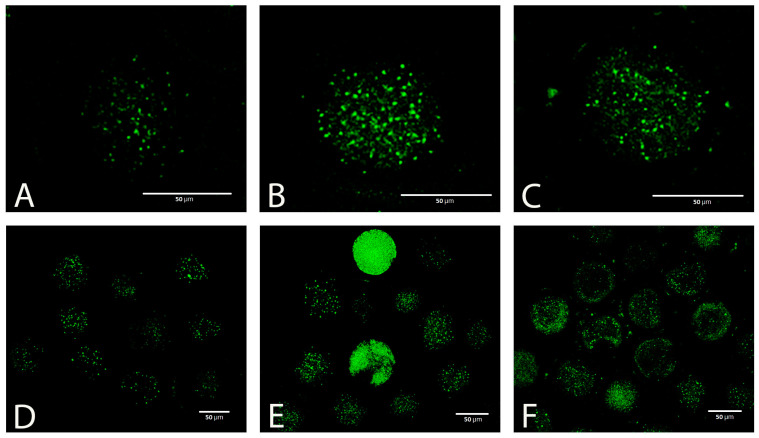
ROS (CellROX green) in fresh (**A**,**D**) and vitrified oocytes without (**B**,**E**) or with (**C**,**F**) GSH supplementation.

**Figure 6 antioxidants-12-00035-f006:**
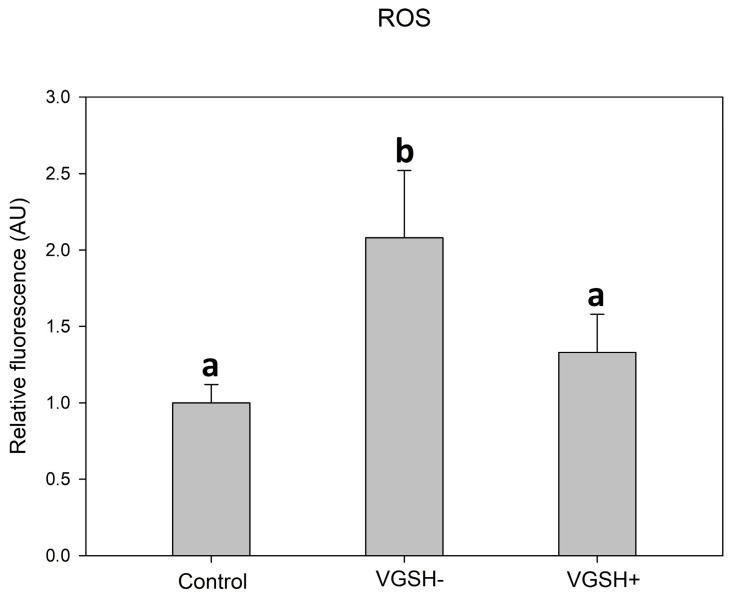
Effect of vitrification and GSH (5 mmol L^−1^) in the post-warming recovery medium on ROS relative fluorescence intensity. Values of relative fluorescence intensity are in arbitrary units (AU). Differences among groups are significant at *p* < 0.05.

**Figure 7 antioxidants-12-00035-f007:**
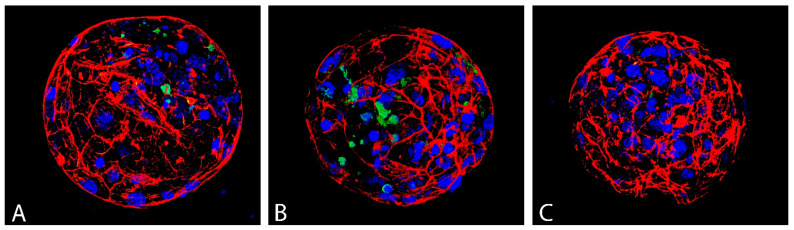
Actin cytoskeleton (red), apoptotic (green) and all nuclei (blue) of D7 bovine blastocyst in control (**A**), vitrified without GSH ((**B**); VGSH−) and with GSH group ((**C**); VGSH+). Confocal microscopy.

**Figure 8 antioxidants-12-00035-f008:**
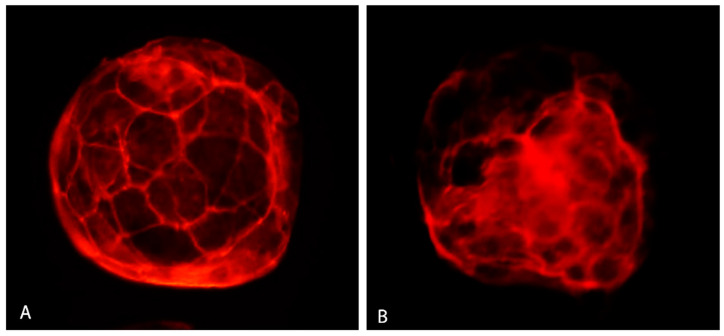
Actin cytoskeleton in bovine IVP blastocyst. Good quality actin (**A**) and poor quality actin (**B**). Fluorescent microscopy.

**Table 1 antioxidants-12-00035-t001:** Effect of vitrification and GSH on embryo development of devitrified oocytes.

Group	Fertilized Oocytes, n	Cleavage Rate, n (%)	Total Blastocyst Rate (TBR), n (%)	D7 Bl from TBR, %
Control	419	279 (66.58)	137 (32.70) ^a^	74.45 ^a^
VGSH−	198	116 (58.58)	40 (20.20) ^b^	55.00 ^b^
VGSH+	223	125 (56.05)	39 (17.49) ^b^	71.79 ^a^

VGSH−—vitrified oocytes without GSH supplementation; VGSH+—vitrified oocytes with GSH (5 mmol L^−1^) supplementation; D7—day 7; Bl—blastocysts. ^a,b^—Values with different superscripts within the columns are significantly different (*p* < 0.05).

**Table 2 antioxidants-12-00035-t002:** Effect of vitrification and GSH on the dynamics of blastocyst formation.

Groups	Total Blastocyst	D6 Bl, n (%)	D7 Bl, n (%)	D8 Bl, n (%)
Control	137	28 (20.44) ^a^	74 (54.01)	35 (25.55) ^a^
VGSH−	40	5 (12.50) ^b^	17 (42.50)	18 (45.00) ^b^
VGSH+	39	9 (23.07) ^a^	19 (48.71)	11 (28.20) ^a^

VGSH−—vitrified oocytes without GSH supplementation; VGSH+—vitrified oocytes with GSH supplementation (5 mmol L^−1^); D6—day 6; D7—day 7; D8—day 8; Bl—blastocysts. ^a,b^—Groups with different superscripts are significantly different in the distribution of blastocyst according to the speed of development (*p* < 0.05).

**Table 3 antioxidants-12-00035-t003:** Total cell number and incidence of apoptotic cells in tested groups.

Group	No. of Blastocysts n	Total Cell Number x ± SEM	TUNEL-Index x ± SEM
Control	88	112.56 ± 3.39 ^a^	11.17 ± 0.58 ^a^
VGSH−	26	98.35 ± 6.25 ^b^	10.68 ± 1.33 ^a^
VGSH+	32	101.00 ± 4.58 ^b^	8.18 ± 0.75 ^b^

VGSH−—vitrified oocytes without GSH supplementation; VGSH+—vitrified oocytes with GSH supplementation (5 mmol L^−1^). ^a,b^—Values with different superscripts in the same column are significantly different (*p* < 0.05).

**Table 4 antioxidants-12-00035-t004:** Distribution of blastocysts according to the quality of actin cytoskeleton.

Group	No. of Blastocysts	Good Quality Actin N (%)	Poor Quality Actin N (%)
Control ^a^	70	50 (71.43)	20 (28.57)
VGSH− ^b^	17	11 (64.70)	6 (35.30)
VGSH+ ^a^	20	15 (75.00)	5 (25.00)

VGSH−—vitrified oocytes without GSH supplementation; VGSH+—vitrified oocytes with GSH supplementation (5.0 mmol L^−1^). ^a,b^—Groups with different superscripts are significantly different in the distribution of embryos to the actin quality grades (*p* < 0.05).

## Data Availability

Data supporting reported results can be found, on: https://docs.google.com/spreadsheets/d/1dnfvY4xsHtXH8rWjsbR9pdpTCpOCq5BHeLDw56jR42w/edit?usp=sharing.

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
