# Peer review of "Glutathione during Post-Thaw Recovery Culture Can Mitigate Deleterious Impact of Vitrification on Bovine Oocytes"

_antioxidants, 2022, doi:10.3390/antiox12010035_

Round 1

Reviewer 1 Report

lines 278-280: "This section may be divided by subheadings. It should provide a concise and precise description of the experimental results, their interpretation, as well as the experimental conclusions that can be drawn": This sentence is not necessary. Please, omit it.

Author Response

Thank you for reviewing the manuscript and pointing out the part of the template that was left out by mistake. This paragraph has been deleted.

Reviewer 2 Report

It is an interesting manuscript that submitted to this journal. Overall, the paper is well-written and the data are well-presented. However, the Abstract is poorly written, suggest that it should be rewritten.

Author Response

Thank you for reviewing the article and for pointing out the weak side of the manuscript. We tried to improve the quality of the abstract. We tried to write the abstract as best as possible, unfortunately the magazine's limited range of 200 characters forced us to drastically shorten it, which affected its quality.

Reviewer 3 Report

Review report

Glutathione during post-thaw recovery culture can mitigate 2 deleterious impact of vitrification on bovine oocytes by Olexiková et al.

The manuscript describes experiments that suggest that the presence of glutathione can improve the quality of vitrified bovine oocytes during thawing.

The experiments are well described and the results presented accordingly. At places the conclusions are not entirely supported by the results. Minor editing for the English language is suggested.

Major comments

-Paragraph 2.1 is redundant, as all techniques are further described.

-Paragraph 3.1 . Please explain in 1 sentence what is the purpose, experiment. Also explain abbreviation VGSH.

Figure legends of Figure 1: Explain the differences between the panels. Explain the y axes nkat/ml. Panel Glutathione reductase: 0,5 should be 0.5 etc.

Lines 273-280: This text is for the instructions to authors, please remove.

Line 284: the decrease in intensity of mitochondrial activity (figure 2)  cannot be compared with the decrease in lysosomal intensity. These are totally different measurement, one cannot say that the decrease of a is more pronounced than that of b. Why is the presence of glutathione not examined here. Data on these readouts with glutathione at different concentrations seems relevant for the conclusion that glutathione exposure increases the oocyte competence.

Figure 3: What re the yellow dots I B. Why is there no mitochondrial staining in the cumulus cells in B? The absence of mitochondrial staining in B suggests different settings for this panel, and reduces credibility of the data.

How was mitochondrial distribution quantified (Fig 3). This should have been done double blinded by the observer(s), or using unbiased software.

Figure 4: the colocalization is not convincing. This should be verified by higher resolution/ magnification and ideally different techniques. Similarly, the conclusion in line 438 is too strong. Colocalization is not convincingly demonstrated.

Figure 5: Assuming that panels A,B,C are of different magnifications than D, E, F, please introduce scale bars. On the y axis: 0.0; 0.5 etc instead of 0,0; 0,5 etc.

Table 2: How were these data obtained? Double blinded? How was objectivity secured?

Line 376-377 is not clear

Table 4: How were these data obtained? Double blinded? How was objectivity secured?

In the discussion line 428 it is suggested that mitochondrial activity was decreased. However, in this study mitochondrial activity was not specifically determined. Instead, mitochondria distribution was determined. The conclusion should therefore be changed.

Line 447-488 is too speculative. Please remove.

Line 573-575: this is part of the instructions for authors. Please remove.

Minor comments

-Reference 49 may not be the most optimal as it is in Polish and not accessible for everybody.

-Reference 52 journal name in italics.

-Line 70 similar to instead of similarly as

-Line 72: increase or improve instead of induce

-Line 106 using a sterile 5mL

-Line 108 in instead of into

-Line 109 with holding medium instead of with a holding madium

-Line 121 off excessive instead of off an excessive

-Line 123 10% FBS instead of 10% of FBS

-Line 127 micropipette; excessive instead of micropipette; an excessive

-Lines 187, 203, 237: confocal laser scanning microspore instead of laser scanning confocal microscope

-Line 347 lower instead of less

-Line 410 as compared to fresh embryos

-Line 414 membrane instead of membranes

There are various line that contain tautologies. For example line 483: could probably; 489 may probably; 503 Possible…may be probably.  Please change.  

Author Response

Thank you for thoroughly revising our manuscript, which will contribute to improving its quality and removing errors.

Major comments

- Paragraph 2.1 is redundant, as all techniques are further described.

Answer: Section 2.1 tries to indicate the scheme and sequence of the individual experiments performed, therefore we included it at the beginning of the description of the methodology. Although the individual methodologies used are subsequently thoroughly described, in our opinion, it is significant because more than one independent series of experiments was performed.

- Paragraph 3.1 . Please explain in 1 sentence what is the purpose, experiment. Also explain abbreviation VGSH.

Answer: A sentence was added to the text of paragraph 3.1 to explain the purpose of the analysis.

„In order to determine the effect of vitrification and the addition of GSH after devitrification on important components of oocyte antioxidant protection, we investigated the total antioxidant capacity, activity of catalase, peroxidase and glutathione reductase in lysed cumulus-oocyte complexes.“

The abbreviation VGSH is explained below the graph. An explanation has also been added to the text.

- Figure legends of Figure 1: Explain the differences between the panels. Explain the y axes nkat/ml. Panel Glutathione reductase: 0,5 should be 0.5 etc.

 Answer: The significance of the differences is described in the figure legend located below the figure. The katal (symbol: kat) is the unit of catalytic activity in the International System of Units (SI) used for quantifying the catalytic activity of enzymes and other catalysts. Unit nkat (nanokatal) is derivated from katal. Since kat belongs to the SI system of units, it does not need further explanation.             On panel Glutathione reductase commas have been replaced by dots.

- Lines 273-280: This text is for the instructions to authors, please remove.

Answer: The text has been removed.

- Line 284: the decrease in intensity of mitochondrial activity (figure 2)  cannot be compared with the decrease in lysosomal intensity. These are totally different measurement, one cannot say that the decrease of a is more pronounced than that of b.

Even if the sentence sounded like that, it was not our effort to compare the results of measuring the fluorescence intensity of MitoTracker green and LysoTracker with each other. The sentence simply says that there was also significant decrease in the intensity of LysoTracker fluorescence. The sentence has been edited.

- Why is the presence of glutathione not examined here. Data on these readouts with glutathione at different concentrations seems relevant for the conclusion that glutathione exposure increases the oocyte competence.

Answer: The aim of measuring the fluorescence intensity of MitoTracker green staining was to observe how mitochondria react to vitrification/warming, because, first of all, mitochondria are considered to be the main source of free oxygen radicals and therefore oxidative stress.(Based on our results, this suggests that other mechanisms may probably be involved in oxidative stres, as we found an increased intensity in ROS staining but MitoTracker showed a lower intensity in oocytes after vitrification/warming. Mitochondria opositely shoved some deficiency after vitrification. This is further discussed in the article.) The aim of the experiment was only to demonstrate the effect of vitrification on oocyte mitochondria. Therefore, the effect of GSH addition was not investigated in these experiments. To verify the impact of GSH on oocyte competence, the assessment of developmental competence after their in vitro fertilization is best demonstrated. The graphs have been modified.

- Figure 3: What re the yellow dots I B. Why is there no mitochondrial staining in the cumulus cells in B? The absence of mitochondrial staining in B suggests different settings for this panel, and reduces credibility of the data.

Answer: I don't know what exactly was meant by the yellow dots, but the problem probably lies in the fact that with 3A, an image with all channels, including red, was mistakenly used. By overlaying red and green in some structures, yellow to orange markings are created. The image has been replaced with the same one containing only the blue and green channels.

We tried to acquire images with the same microscope settings. However, properties of the individual microscopic preparation sometimes change the optical environment and cause changes in the images. Mitochondrial staining in cumulus cells in 3B  is indeed not visible in this image, even though the images were acquired at the same settings. In my experience this is due to worse claening of the oocyte. The final image is obtained by overlapping several optical sections taken through the oocyte. If the oocyte is poorly devoid of cumular cells, their nuclei overlap in a continuous layer and form too intense blue fluorescence around the oocyte, therefore the green fluorescence of mitochondria disappears in these places. Whereas in image A and C, only a few individual cells remain (they were better cleaned) and the mitochondrial staining is therefore clearly visible.

A representative image has been replaced with another with a smaller cumulus cell remnant.

- How was mitochondrial distribution quantified (Fig 3). This should have been done double blinded by the observer(s), or using unbiased software.

Answer: Classification into one of the three distribution patterns was done by a qualified observer. I don't know if the term "double-blind" can be used in the case of our experiments, which is used primarily in clinical trials. The practice in our laboratory, to eliminate the risk of subjectively influencing the results, is such that one researcher provides fluorescent staining and mounting of the preparations and another experienced researcher provides microscopy photography of the numbered preparations and evaluations.

- Figure 4: the colocalization is not convincing. This should be verified by higher resolution/ magnification and ideally different techniques. Similarly, the conclusion in line 438 is too strong. Colocalization is not convincingly demonstrated.

Answer: I agree that higher resolution imaging methods, possibly electron microscopy, would provide more valuable evidence. However, the overlap of red and green fluorescence also indicates to some extent that the occurrence of these organelles is often in close proximity. Claims that we have clear evidence of colocalization were corrected in the article.

- Figure 5: Assuming that panels A,B,C are of different magnifications than D, E, F, please introduce scale bars. On the y axis: 0.0; 0.5 etc instead of 0,0; 0,5 etc.

Answer: Figures ABC are details on one oocyte from group and DEF are group figures of oocytes. Scale bares were added.

On the axis on panel ROS commas have been replaced by dots.

- Table 2: How were these data obtained? Double blinded? How was objectivity secured?

 Answer: Evaluation of the development of pre-implantation embryos is based on the observation of morphological features. During the assessment of blastocysts, we focus primarily on observing the formation of the blastocoel cavity. The evaluation is performed by an experienced observer according to clear criteria defined by the International Embryo Transfer Society (IETS). This morphological evaluation brings with it a certain degree of subjectivity, however, the use of international IETS criteria and the experience of the observer guarantee the correctness of the evaluation. Such evaluation is common practice in assessing the development of preimplantation embryos.

- Line 376-377 is not clear

 Sentences were rewritten.

- Table 4: How were these data obtained? Double blinded? How was objectivity secured?

 Answer: Classification into one of the three distribution patterns was done by a qualified observer. The practice in our laboratory, in florescent staining and microscopic evaluation, to eliminate the risk of subjectively influencing the results, is such that one researcher provides fluorescent staining and mounting of the preparations and another experienced researcher provides microscopy and photography of the numbered preparations and their evaluations according to predetermined criteria (see on representative figures Fig. 8 AB).

- In the discussion line 428 it is suggested that mitochondrial activity was decreased. However, in this study mitochondrial activity was not specifically determined. Instead, mitochondria distribution was determined. The conclusion should therefore be changed.

Answer: In our study, in addition to distribution, the activity of mitochondria was measured indirectly by measuring the intensity of fluorescence in individual oocytes after staining with MitoTracker Green. The results of the measurements are presented in Figure 2. The method is based on fluorescence quantification of the mitochondrial mass of active accumulating mitochondria. It was in oocytes established and described by the authors Angello, Morici and Rinaldi; 2008. MitoTracker™ Green is non-fluorescent in aqueous solutions, becomes fluorescent only when it is active accumulated in the mitochondrial lipid environment, regardless of membrane potential. So the measured fluorescence intensity is dependent on the active accumulation of mitochondria and their mass, regardless of the membrane potential. This method, although indirectly, tells about the presence and the amount of active mitochondria in the oocyte.

Based on our measurement results and statistical assessment of the difference (Results Fig.2), we claim that the occurrence of active mitochondria in oocytes after vitrification is significantly lower.

- Line 447-488 is too speculative. Please remove.

Answer: Were paragraphs in the range 447 to 488 really meant? Or maybe it's lines 447-448? The statement in lines 447-448 has been modified, in the sense that it is only one of the considered possible hypotheses, but it needs further investigation.

- Line 573-575: this is part of the instructions for authors. Please remove.

Answer: Part was removed, thanks for warning.

Minor comments

-Reference 49 may not be the most optimal as it is in Polish and not accessible for everybody.

-Reference 52 journal name in italics.

-Line 70 similar to instead of similarly as

-Line 72: increase or improve instead of induce

-Line 106 using a sterile 5mL

-Line 108 in instead of into

-Line 109 with holding medium instead of with a holding madium

-Line 121 off excessive instead of off an excessive

-Line 123 10% FBS instead of 10% of FBS

-Line 127 micropipette; excessive instead of micropipette; an excessive

-Lines 187, 203, 237: confocal laser scanning microspore instead of laser scanning confocal microscope

-Line 347 lower instead of less

-Line 410 as compared to fresh embryos

-Line 414 membrane instead of membranes

-There are various line that contain tautologies. For example line 483: could probably; 489 may probably; 503 Possible…may be probably.  Please change.

Answer: Reference 49 has been replaced by another article proving our claims, in the English language. All other minor comments were taken into account and corrected in the article.  

Reviewer 4 Report

The study provides additional information on the role of antioxidants (glutathione, GSH) in protection of vitrified-rewarmed bovine oocytes. Addition of GSH to the recovery medium proved to mitigate oxidative stress (OS)-related cellular damage and to improve developmental potential of the vitrified/warmed oocytes. The study appears to have important clinical implications and may serve as a basis for further research.

The reviewer has some suggestions/questions for consideration:

-          High dose GSH (5 mmol/l) in the recovery medium increased the total antioxidant capacity (TAC) of lysed oocytes. As TAC includes only non-enzymatic antioxidant elements the question may arise whether GSH-induced increase in TAC is due to GSH in itself or other elements are also  involved?

-          The responses of antioxidant enzyme acitivities to vitrification/rewarming are inconsistent. It is reasonable to assume, therefore, that simultaneous determination of their mRNA expression and protein levels would provide more convincing result.

-          The lines 278-280 can be deleted.

Author Response

-          High dose GSH (5 mmol/l) in the recovery medium increased the total antioxidant capacity (TAC) of lysed oocytes. As TAC includes only non-enzymatic antioxidant elements the question may arise whether GSH-induced increase in TAC is due to GSH in itself or other elements are also  involved?

Answer: Thank you for reviewing our manuscript and for your encouraging review. By investigating enzymatic antioxidant systems, we just wanted to get closer to answering this question. It seems that the answer will not be simple, while in one part of the whole system we detected an increase in activity (peroxidase) in another part, on the contrary, a decrease (catalase). In any case, it seems that the addition of non-enzymatic GSH to the medium after oocyte thawing will affect in a certain way also the enzymatic systems of antioxidant protection.

-          The responses of antioxidant enzyme acitivities to vitrification/rewarming are inconsistent. It is reasonable to assume, therefore, that simultaneous determination of their mRNA expression and protein levels would provide more convincing result.

Answer: It's an interesting proposition. It is somewhat problematic that the oocytes at the time of vitrification are in the sensitive phase of entering meiosis. The ability to respond at this time to any stress by increased mRNA expression is limited. Overall, the restoration of this ability is attributed to pre-implantation embryos after overcoming the maternal-zygotic transition, at the stage of 8-16 cells after fertilization of the oocyte. However, our further research is directed to the question of whether vitrification and the addition of an antioxidant after varming the oocyte can manifest itself in the mRNA expression of important genes for antioxidant protection as well as other developmentally relevant genes in the blastocyst stage.

-          The lines 278-280 can be deleted.

Answer: Thank you for pointing out the part of the template that was left out by mistake. This paragraph has been deleted.

Round 2

Reviewer 3 Report

Most of the comments I made to the previous manuscript have been satisfactorily answered/ dealt with.